# Effects of Electropolishing on Mechanical Properties and Bio-Corrosion of Ti6Al4V Fabricated by Electron Beam Melting Additive Manufacturing

**DOI:** 10.3390/ma12091466

**Published:** 2019-05-07

**Authors:** Yao-Cheng Wu, Che-Nan Kuo, Yueh-Chun Chung, Chee-How Ng, Jacob C. Huang

**Affiliations:** 1Department of Materials and Optoelectronic Science, National Sun Yat-Sen University, Kaohsiung 804, Taiwan; c8921570@gmail.com (Y.-C.W.); book12343456@gmail.com (Y.-C.C.); cheehow999@gmail.com (C.-H.N.); 2Department of Bioinformatics and Medical Engineering, Asia University, Taichung 413, Taiwan; cnkuo@asia.edu.tw; 33D Printing Medical Research Institute, Asia University, Taichung 413, Taiwan; 4Institute for Advanced Study, Department of Materials Science & Engineering, City University of Hong Kong, Kowloon, Hong Kong, China

**Keywords:** electron beam melting, surface roughness, Ti6Al4V, mechanical property, electrochemical analysis

## Abstract

Electron beam melting (EBM) has become one of the most promising additive manufacturing (AM) technologies. However, EBM tends to result in products with rougher surfaces due to the melt pool which causes adjacent powder particles to be sintered to the surface without being melted. Hence, it is necessary to improve the surface quality by post processing. The current study evaluates the tensile response of Ti6Al4V EBMed samples subject to various electropolishing (EP) treatments. The surface roughness Ra readings can be improved from over 24 µm down to about 4.5 µm by proper EP, resulting in apparent tensile elongation improvement from 7.6% to 11.6%, or a tensile plasticity increment of 53%, without any loss of elastic modulus or tensile strength. Moreover, the in-vitro bio-corrosion test in simulating body fluid (SBF) of the as-EBMed and EP-processed samples is also conducted. The potentiodynamic polarization reveals that the bio-corrosion resistance is improved by the lower Ra through proper EP treatments. This is due to the formation of a denser and more completely passivated oxide layer with less defects after proper EP duration. But when the EBMed samples are over-electropolished, nano pitting would induce a degraded bio-corrosion performance.

## 1. Introduction

Additive manufacturing (AM), also known as three-dimensional printing (3-D printing), is a technology which is used to fabricate objects directly from a 3D model and print layers by layers [1,2]. Each layer is a cross-section of the work piece derived from the Computer Aided Design (CAD) data. Additionally, it can be classified into many types depending on what materials are used, how the layers are built, and how the layers are bonded to each other. These major differences will influence mechanical properties and the accuracy between the CAD models and final products. For metal powder materials, powder bed fusion (PBF) is the most promising 3D printing technology [3]. Electron beam melting (EBM), one kind of PBF technology, is conceptualized and patented by Arcam AB^®^ based in Sweden [4]. The process uses the electron beam to melt metal powders in a vacuum chamber [5]. Due to the preheating steps, an elevated temperature of about 700 °C is maintained in the chamber to reduce residual stresses [6].

However, as-fabricated objects by EBM meet the issue of surface roughness, which is mainly caused by powder particles sticking to the molten surface contour during manufacturing [7]. Besides discussing the effect of powder particles, Vandenbroucke et al. first showed that the conditions of materials, layer thickness, scanning parameters and scanning strategy also strongly influence the outcome of surface roughness [8]. Since the roughness might cause stress concentration and crack initiation, it also has a negative impact on the fatigue performance [9,10,11,12]. Therefore, the surface quality has to be improved by suitable post processing. Milling and blasting are well known post processing methods that are applied for additive manufacturing built-parts [13,14]. However, milling might not be suitable for complex geometries which are undercuts or inner structures. On the other hand, blasting might be applicable on more complicated surfaces, but both of them are mechanical processes which modify surface roughness by direct physical contact. Moreover, as described in study of Urlea et al. electropolishing is advantageous to eliminate surface features of AM products when compared to chemical polishing treatment [15]. Lots of studies indicate that surface roughness is successfully reduced after an electropolishing treatment. Han et al. [16] investigated the corrosion rate of 316L stainless steel after electropolishing within deaerated high temperature water. Corrosion resistance of Ti-15Mo alloy after electropolishing and anodic oxidation were discussed in the work by Babilas et al. [17]. Therefore, this study investigates the effect of the electropolishing (EP) process, which might improve the surface quality and reduce the surface roughness Ra readings without direct physical contact. 

Electropolishing (EP) is an electrochemical process in which the corrosion of an anode, resting in an electrolyte, is accelerated by the introduction of an electric current into an electrolytic cell. There are a number of primary components in an electrolytic cell, an anode, a cathode, a power supply, and an electrolyte accommodating the reaction [18]. In an electrolytic cell, the electropolishing of the anode and the electroplating of the cathode occur at the same time. As current passes through the sample, the protrusion of the surface will concentrate the electric field and accelerate the corrosion, causing electropolishing to initially reduce the surface locally at larger features [19]. On the other hand, when metal ions diffuse into electrolyte and reach saturation, the sample surface will form a metastable polishing film, which has high resistance and viscosity. If the polishing film becomes too thick, the diffusion of metal ion will be inhibited, and influence the polishing. Therefore, keeping a proper thickness of polishing film can make sure the ion diffusion develops the leveling and smoothing effect [20].

Past research has have demonstrated that Ti-based alloy is one of the most promising bio-implant materials and has been widely applied in orthopedic replacement and dental implantation [21,22]. In particular, the properties of superior corrosion resistance, small density, relatively low elastic modulus and favorable mechanical strength are the advantages of Ti-based alloys [23,24,25,26]. Moreover, Ti-based alloys have been one of the main materials that are fabricated by EBM. According to some of the literature, the appropriate pore size helps bone ingrowth, and the appropriate surface roughness helps cell adhesion [27,28]. There are also many issues that were investigated before [29]. Although implants with appropriate surface roughness are positive to cell adhesion, poor surface roughness would lead to poor bio-corrosion resistance [30]. In addition, the application of additive manufactured products for bio-implantation is suspicious due to inferior surface properties such as high surface roughness or micro-cracks caused by adhesion of unmelt powder particles, the heat gradient (balling effect) or the entrapment of gases. Walter et al. [31] point out that as surface roughness decreases, the passivation and corrosion behaviors of AZ91 magnesium alloy such as corrosion potential, corrosion current density and pitting resistance increase. Zhang et al. [32] conducted electrochemical tests through electropolished Ti-based alloy fabricated by selective laser melting within Ringer’s solution, different electrochemical behaviors were shown under different surface roughness or polishing extents of sample. Since the poor surface properties are doubted in deteriorating the subsequent mechanical response and bio-corrosion resistance, the as-EBMed Ti6Al4V is optimized in this study by using the EP method.

## 2. Material and Methods

The Ti6Al4V powders fabricated by plasma atomization technology were purchased from Arcam AB^®^, Mölndal, Sweden. The AM EBM process was conducted by Arcam Q10 system which was developed as the 3rd generation EBM technology. During the EBM process, a partial pressure of He is introduced to a 1 × 10^−3^ mbar, with a current of 15–17 mA, scan speed of 4.5 m/s, beam diameter of 100 µm, hatch spacing of 150 µm and 50 µm thickness of the single layer. Materialise Magics CAD software (19.0, Materialise, Leuven, Belgium) was used to design the CAD models of tensile test specimens with a gage section of 16 mm in length, 3.2 mm in width, and 1.5 mm in thickness, as shown in Figure 1. After the EBM process, the specimens were buried in the un-melted powders. The Arcam Powder Recovery System (PRS), using a high pressure system to eject the Ti-6Al-4V powders, is adopted to take away the ill-sintered powders from the sample surface.

The ring-shaped Ti6Al4V cathode was designed in this experiment and the schematic sketch setup is shown in Figure 2. The sample is set at the center of ring-shaped cathode in order to get uniform polishing because the distance from each surface of sample to the cathode is the same. The electrolyte chosen for EP experiments was composed of acetic acid, 60% perchloric acid, and 95% ethanol, mixed in a 16:5:4 vol ratio. Before the EP treatment, Ti6Al4V samples were sequentially cleaned by ultrasonic vibration in acetone and ethanol. During EP, the sample (anode) was immersed in electrolyte, and the current (usually DC) was connected between the sample and the cathode. The glass beaker was used as container of electrolyte. A magnetic stirring bar was placed at the bottom of the beaker to provide agitation during EP. The electrolyte was agitated at a constant speed of 800 rpm using magnetic stirring bar rotated by coring stirring hot plate. During the experiment, the beaker was placed in an ice water bath in order to maintain low temperature.

The appearances of atomized powders were firstly examined by scanning JOEL JSM-6330 (Tokyo, Japan), and the quantitative image analysis was used by ImageJ software (1.4, Wayne Rasband, Bethesda, MD, USA) to characterize the powders distribution and pore size. Scanning electron microscopy (SEM) was also applied to observe the microstructure and surface morphology of samples before and after electropolishing. The JEOL JAMP-9500F X-ray photoemission spectroscopy (XPS, Tokyo, Japan) is used to analyze the chemical compositions of surface before and after EP. The XPS with Mg–Ka (1253.6 eV) radiation was operated at 10 kV and 5 mA under the vacuum pressure of 10^−7^ Pa. The binding energy peaks of each element could be presented by wide-scan and narrow-scan, and they were calibrated by the carbon C-1s peak.

Veeco Dektak 150 Stylus Profiler 3D alpha-step profilometer (New York, NY, USA) was applied to determine the surface roughness of specimens. A nano-probe with stylus force of 5 mgf is given during scan to identify the height differences on surface. Every scan operated under acquisition rate of 6.5 µm/s, length of 2 mm and scanning rate of 1000 µm per minute. The average value of surface roughness was obtained after a level correction. All the specimens including the as-EBMed and EPed ones were measured at least 10 times.

Before tensile testing, the gauge length (16 mm), width (3 mm) and thickness (1.5 mm) of each tensile sample need to be carefully measured and marked. Afterwards, the specimens were subjected to tensile testing with a strain rate of 1 × 10^−3^ s^−1^ at room temperature by using the Instron 5582 universal testing machine (Norwood, MA, USA). All tensile tests were conducted for at least three times, and the averages were presented. 

Since the EBMed Ti6Al4V products were designed for bio-implant applications, the electrochemical bio-corrosion performance was assessed by various kinds of electrochemical tests. A commercial electrochemical analyzer system, CHI 614 D (CH Instruments Inc., Austin, TX, USA), was used under a three-electrode system scheme. The working, reference and counter electrode were Ti6Al4V sample with around 16 mm^2^, platinum film and a standardize Ag/AgCl electrode, respectively. The simulated body fluid (SBF) with concentration of 0.137 M of NaCl, 5.4 mM of KCl, 4.2 mM of NaHCO_3_, 1.0 mM of MgSO_4_, 0.44 mM of KH_2_PO_4_, 1.3 mM of CaCl_2_, 0.25 mM of Na_2_HPO_4_, pH = 7.4 was introduced as the media. Before starting potentiodynamic polarization measurements, the three electrodes were immersed into Hank’s solution until the criteria was achieved as a variation of open circuit potential (OCP) within 2 mV in 10 min. The polarization scan started from the readings of OCP substrate 0.2 V, and ended up at 2 V with a scanning rate of 0.33 mV/s.

All the raw data was collected by the devices including 3D alpha-step profilometer, universal testing machine and electrochemical analyzer system, and was exported as excel files. The software used for analysis and statistics is OriginPro 8 (OriginLab, Northampton, MA, USA) which is applied to draw the figures in this study. All the feature points are obtained from these figures.

## 3. Results and Discussions

### 3.1. Powder Analysis

The as-received Ti6Al4V powders practically appear smooth, spherical, fully dense and sometimes with a few non-spherical powders, as shown in Figure 3a. Using the image analysis software ImageJ, the circularity of the Ti6Al4V powders was about 0.962 ± 0.024, which was considered to be high circularity, beneficial for powder flow during AM EBM [33]. The average powder size (d_50_) is 75.1 ± 0.8 µm, determined from the particle size distribution shown in Figure 3b measured by the particle size analyzer (Mastersizer 2000, Malvern, Worcestershire, UK).

### 3.2. Surface Roughness

Figure 1 indicates where the lateral surface and upper surface are. The lateral surface roughness is a result mainly affected by the parameters of the contour melting; the parameters are the same for each layer during processing, so that the surface roughness of lateral surface would be similar. On the other hand, the surface roughness of the upper surface is often different because various positions or shapes may adopt different scanning strategies [34]. Therefore, the lateral surfaces were used to characterize the relationship between surface roughness and tensile/bio-corrosion responses in this study. The anodic current density (J) versus voltage plot for the EP processing is shown in Figure 4. It can be seen that there is no current density limit plateau. Therefore, the current density is the rate determining process of the EP process [35]. The surface roughness Ra evolution as a function of EP time is illustrated in Figure 5. The average roughness of the as-EBMed sample, namely, the unpolished sample, is 24.1 ± 2.6 µm. Figure 5a depicts the decreased surface roughness Ra data under the condition of a constant current density of 147, 294, and 442 mA/cm^2^. The sample electropolished at 147, 294 and 442 mA/cm^2^ for 20 min can reach a surface roughness level about 15.1 ± 1.2, 10.2 ± 0.9, and 4.5 ± 0.5 µm, respectively, as shown in Table 1. Parallel testing has also been conducted for the EBM samples under the constant voltage condition, as one example shown in Figure 5b, attaining a Ra level of 19.0 ± 0.7, 14.9 ± 0.6, 10.4 ± 1.2 and 4.4 ± 0.8 µm with EP duration of about 3, 7, 13 and 20 min, respectively. It can be seen that, with a proper EP treatment, either under constant current density or constant voltage, the surface roughness can be effectively improved from over 20 µm down to below 5 µm.

### 3.3. Surface Morphology

Figure 6a–d shows the SEM morphology with different degrees of surface roughness. As shown Figure 6a, the as-fabricated EBM sample shows poor surface quality as a large quantity of spherical powders bonded all over the surface. All of the as-EBMed specimens have been treated by PRS. Basically, the ill-sintered powders have been removed. The particles in the SEM image in Figure 6a are those partially melted particles, which are difficult to removed using PRS. Although those powders showed comparatively weak bonding or incomplete fusion, they were still unable to be cleaned up through supersonic vibration, and became the major factor causing high surface roughness of additive manufactured samples. 

In the early stage of the EP process, those partially melted powders are firstly removed, and the surface morphology then becomes the SEM image of Figure 6b. Although there are ups and downs on the surface after removing the particles, the surface roughness is not really rougher than the as-fabricated one. Figure 6b–d show the surface morphology of EBM samples subject to EP treatment under the constant current density condition at 147, 294, and 442 mA/cm^2^ and 4 °C for 20 min. These three samples possess surface roughness levels of about 15 µm, 10 µm and 4.5 µm. Since EP has be introduced for 20 min, the apparent spherical-like powders attached on surface were gradually eliminated from the as-EBMed sample surface. Then, the surface was transformed into a field-like morphology, when the surface roughness was lowered down to about 10 µm. When the Ra reading is reached to 4.5 µm, the overall surface basically displays a smooth surface examined at lower magnifications. However, careful examination at a much higher magnification showed that the samples with the lowest Ra of 4.5 µm started to exhibit some small black dots, as one example presented in Figure 6e shows. Such tiny black dots, 1 µm or less in size, appeared to be a sign of over- electropolishing. 

### 3.4. Surface Compositions Analysis

In order to realize the surface compositions before and after EP, X-ray photoemission spectroscopy (XPS) was introduced to analyze the surface, as shown in Figure 7. The wide-scan XPS spectra reveal the Ti, Al, V and O peaks. The narrow-scan of each element is presented in Figure 7a–d, respectively. In Figure 7a, the two separated peaks at about 464.1 and 458.2 eV correspond to the Ti^4+^ state from TiO_2_ [36]. The two separated peaks in Figure 7b at about 74.8 and 71.9 eV are contributed to by Al^3+^ state from Al_2_O_3_ and Al^0^ state [37]. The single peak in Figure 7c at 516.9 eV is caused by V^5+^ state from V_2_O_5_ [36]. The oxygen spectrum shown in Figure 7d reveals that the board peak is composed by the oxygen–metal bonding and oxygen–hydrogen bonding peaks. The peak at about 531.6 eV is contributed to by oxygen–hydrogen bonding and Al–O bonding, and the peak at about 529.9 eV is dominated by Ti–O and V–O bonding [38].

Thus, the results indicate that the compound oxide layers are made of TiO_2_, Al_2_O_3_ and V_2_O_5_. According to the XPS results of surface oxide layer, the samples before the EP process (the as-EBMed samples) tend to have a weaker signal accumulation because the surface roughness affects the collection of signals. Moreover, Figure 7d indicates that the samples after EP have a relatively high peak of Ti–O bonding. Thus, this result demonstrated that the EPed samples show the advantages in generating a more homogeneous TiO_2_ layer with less defects, beneficial to the subsequent bio-corrosion response. 

### 3.5. Mechanical Response

To investigate the mechanical properties of EBMed samples subjected to the different degrees of EP, systematic tensile tests were conducted. The resulting tensile stress-strain curves are shown in Figure 8. The mechanical properties with Young’s modulus, yield stress, ultimate tensile strength, and elongation are all listed in Table 1. Firstly, the average Young’s modulus of the unpolished samples with Ra ~ 24 µm, and EPed samples with Ra ~ 15, 10, and 4.5 µm are 100 ± 2, 103 ± 2, 100 ± 1, and 102 ± 3 GPa, respectively. The difference between the unpolished and polished samples is only about 3%, nearly unaffected by EP. This means that the elastic modulus would be independent of sample surface roughness, which is scientifically logical. Secondly, the average yield stress (YS) of the four samples are 813 ± 3, 817 ± 2, 809 ± 5, and 817 ± 2 MPa. Again, there is little increment or decrement of the yield stress caused by EP. The unaffected elastic modulus and yield stress suggest that EP treatment and Ra reduction would not affect the nature of the elastic deformation. Thirdly, the average ultimate tensile stress (UTS) of the four samples was found to be 995 ± 8, 1012 ± 13, 1025 ± 7, 1052 ± 8 MPa, respectively. Gradual improvement of UTS is evident. The increment from 995 to 1052 MPa is about 6%. The elastic modulus and YS are not strongly affected by EP and roughness Ra. But the plasticity and fracture related UTS can be upgraded by 6%. 

In addition, the tensile elongation of the four samples from Table 1 are 7.6 ± 0.4, 8.7 ± 0.5, 9.3 ± 0.4, and 11.6 ± 0.7%. The EP-polished samples with the lowest Ra of 4.5 µm appear to exhibit tensile elongation better than that of the unpolished one by 53% (improved from 7.6% up to about 11.6%). The reason can be explained by the different roughness levels for the side surfaces of tensile specimens. The surface roughness serves as the pre-crack, which results in the stress concentration. The stress concentration causes a localized increase in stress. The maximum stress (σ_max_) attributed to stress concentration from surface roughness can be calculated by the formula [39]
(1)$$\sigma_{\max}=\sigma \left(1+2\sqrt{\frac{d}{\rho }}\right)$$
where σ is the uniform load applied on the specimens, d is the depth of the notch and ρ is the radius of the notch. The maximum stress leads to localized deformation and propagation of crack. Thus, necking occurs in advance during the tensile testing, and the tensile elongation decreases. In order to avoid the stress concentration, electropolishing was used to reduce the depth of the notch (d) and increase the radius of the notch (ρ).

In a previous study, the ratio of the surface roughness and strut diameter could affect the fracturing work [40]. Since the tip of rough surface could be considered as the pre-crack of the strut, the thinner the strut diameter of the larger the depth of the pre-crack could enlarge the effect of stress concentration. Hence, the fracturing work would be increased by increasing the depth of the pre-crack and thus it is harmful for the toughness of the materials.

Moreover, the stress concentration factor (K_t_) is used to quantify how concentrated the stress is in a material, and it is calculated by the formula [39]
(2)$$K_{t}=1+2\sqrt{\frac{d}{\rho }}$$

Based on Equation (2), K_t_ values of different degrees of notches are calculated and collected in Table 1. From Figure 8, necking occurred earliest on the unpolished sample with Ra~24 µm, but necking occurred on the EPed samples slowly, especially the one with 4.5 µm, as marked by arrows with different colors. Furthermore, it can be observed that the tensile elongation increases from 7.6% to 11.6% with the value of K_t_ decreasing from 2.8 down to 1.2. Therefore, the appreciably improved tensile elongations for the polished tensile specimens seem to be a result of the significant reduction of stress concentration. 

### 3.6. Bio-Corrosion Electrochemical Response

For long-term bio-implant applications, the bio-corrosion behavior in SBF must be examined as an assessment during implantation. The relationship between corrosion potential and immersion time was firstly revealed through open circuit potential (OCP) testing, also known as the E-t test, where E is the OCP corrosion voltage and t is the immerged time. As shown in Figure 9a, all OCP curves of EBMed Ti6Al4V samples with or without EP gradually become smoother with increasing t, indicating the formation of passive film [41,42]. Moreover, the value of corrosion potential (E_corr_) is usually approximately equal to the OCP E reading, these E values are important parameters in determining the opportunity to initiate a polarization reaction. As shown in curves, the OCP value of specimen firstly decreases when an EP treatment is introduced, from −0.25 V of the as-EBMed unpolished sample (Ra ~ 24 µm) down to −0.38 and −0.45 V for Ra ~ 19 and 15 µm, respectively. But the OCP E value of would start to increase while the duration of electropolishing is prolonged, E reading being −0.32 and −0.23 V for Ra ~ 10 and 4.5 µm, as compared in Table 2.

The potentiodynamic polarization Tafel curves of EBM samples with surface roughness about 24, 19, 15, 10, and 4.5 µm are demonstrated in Figure 9b. Certain important corrosion parameters such as corrosion potential (E_corr_), corrosion current density (I_corr_), and pitting corrosion (E_pit_) could be determined from the curve. Firstly, E_corr_ could stand for the activation energy needed for a corrosion reaction involving forming oxides. A system with lower E_corr_ indicates that less energy is needed to activate a polarization oxidation reaction. From Figure 9b, E_corr_ of EBM sample with a surface roughness reading of 24, 19, 15, 10, and 4.5 µm is about −0.270, −0.413, −0.394, −0.369 and −0.225 V, respectively, as listed in Table 2. Basically, similar to the tendency revealed in the OCP curves, the E_corr_ readings of the polished samples firstly decrease and then increase with increasing EP duration and decreasing Ra, as depicted in Figure 10a. These results may have contributed to the outcome that the relatively noble or stable oxide layer initially formed on the sample had been removed by the beginning of the EP treatment. It follows that the polarization oxidation reaction would be initiated earlier from the more negative voltage. But with increasing of the EP time, the oxide layer would be formed again during EP. This would increase the subsequent resistance of polarization oxidation reaction, namely, initiating a polarization oxidation reaction would be more difficult with a less negative voltage. Thus, the variation of E_corr_ as a function of Ra would exhibit a U-shaped trend, as shown in Figure 10a.

From Figure 9b, it seems that the pitting reaction, which occurred in the as-EBMed unpolished samples at 1.85 V, could be suppressed after EP treatment. The pitting potential of all EPed EBM samples is over 2 V. Pitting was not observed in the EPed samples over the polarization voltage range examined (−2 V to +2 V). This result implies that the application of EBMed Ti6Al4V samples with poor surface roughness (without any EP treatment) is suspicious with a potential hazard in SBF at +1.85 V, and would not be considered to be safe for bio-implantation. The passive region, ΔE = E_pit_ − E_corr_, is an index to determine the formation of a stable passive layer during anodic polarization. All EBM samples after EP treatment tested by the Tafel analysis would possess a broad passive region greater than 2 V, suggesting a better and more protective passive oxide layer could be formed. 

In addition, the corrosion current density (I_corr_) revealed from potentiodynamic polarization curves could be realized as the corrosion rate. The corrosion current density of EBM samples with Ra about 24 µm, 19 µm, 15 µm, 10 µm, and 4.5 µm is about 54.36, 29.38, 22.19, 15.47 and 27.24 nA/cm^2^, respectively. The variation of (I_corr_) as a function of Ra reading is presented in Figure 10b. The results indicate that the as-fabricated sample with highest surface roughness of ~24 µm possesses a much higher I_corr_ value (54.36 nA/cm^2^). I_corr_ would gradually decrease with a decreasing Ra, down to 15.47 nA/cm^2^ for Ra ~ 10 µm. In other words, the EP sample with lower surface roughness shows a slower corrosion reaction once the passivated oxide film has partially broken. This variation trend may be realized as a relatively inhomogeneous surface condition, such as grooves and crevice with higher energy level on the rougher surface, would promote the corrosion reaction. However, as shown in Figure 10b, the corresponding I_corr_ value for samples with Ra ~ 4.5 µm increased rather than decreasing with the above decreasing trend. From the SEM micrograph already shown in Figure 6e, the EPed sample with Ra ~ 4.5 µm appears to have been over-electropolished, resulting in some tiny black dots. These local dots would be prone to minor corrosion and even slight pitting during polarization testing, increasing the I_corr_ value up to 27.24 nA/cm^2^. Those over-polished spots would induce inferior effects in promoting bio-corrosion reaction once the passive film is broken.

Lastly, I_pass_ is a parameter applied to estimate whether a protective and denser passive layer has been formed during anodic polarization. The I_pass_ reading of the EBM samples with Ra about 24, 19, 15, 10, and 4.5 µm is 6.71, 2.80, 2.88, 2.68, and 2.90 µA/cm^2^, respectively, as also listed in Table 2. The results again demonstrated that the EPed samples show advantages in generating a more passivate and dense layer in SBF. 

### 3.7. Closing Remarks

Note that the over-electropolishing in this study for the EPed sample with the lowest Ra reading of ~4.5 µm did not seem to cause any negative effect on the mechanical tensile performance. This very EPed sample still exhibits the highest UTS of 1052 MPa and the highest tensile elongation of 11.6%. But the bio-corrosion resistance of this sample has been degraded slightly. Those ~1 µm-sized black dots due to over-EP appear to be too small to cause stress concentration (with the low K_t_ value of 1.2 in Table 1) to degrade the tensile properties, but they are already sufficient to raise local bio-corrosion rates to I_corr_ of 27.24 nA/cm^2^ and I_pass_ of 2.90 µA/cm^2^. Proper EP treatments should result in a flatter surface without tiny EP-induced defects. 

## 4. Conclusions

Based on the results and discussion, the following conclusions can be drawn.
(1)The current spherical, smooth and non-broken shape of Ti6Al4V powders can result in good powder flow and uniform powder bed during EBM. But the as-EBMed surface would contain partially melted powders, forming a rougher surface with Ra ~ 24 μm.(2)The electropolishing treatment is able to achieve surface improvement from Ra over ~24 μm down to about 4.5 μm with proper EP solution, voltage, current density, temperature and duration time.(3)Via the proper EP, the stress concentration effect can be pronouncedly reduced, upgrading the subsequent tensile performance to raise the UTS by 6% and tensile elongation by 53%. With decreasing Ra values, the mechanical response is continuously upgraded in this study.(4)According to potentiodynamic polarization results, the as-EBMed sample with the highest Ra possesses the higher E_corr_, I_corr_ and I_pass_ values. With decreasing Ra, the bio-corrosion rate becomes much lower.(5)However, when the EP treatment is over-done, the tiny dots induced by EP would impose some negative effects on the bio-corrosion in SBF. Proper EP treatments should result in a flatter surface without tiny EP-induced defects.

## Figures and Tables

**Figure 1 materials-12-01466-f001:**
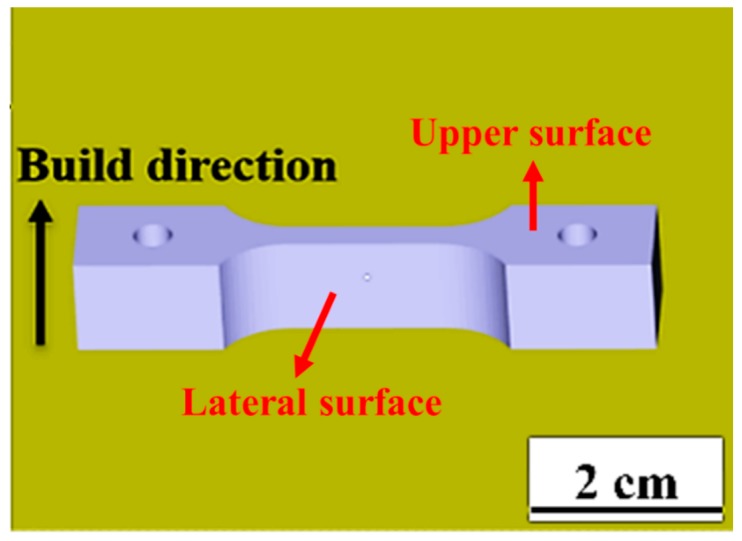
The CAD model is the tensile test specimen with a gage section of 16 mm in length, 3.2 mm in width, and 1.5 mm in thickness. The schematic also indicates the lateral surface and upper surface.

**Figure 2 materials-12-01466-f002:**
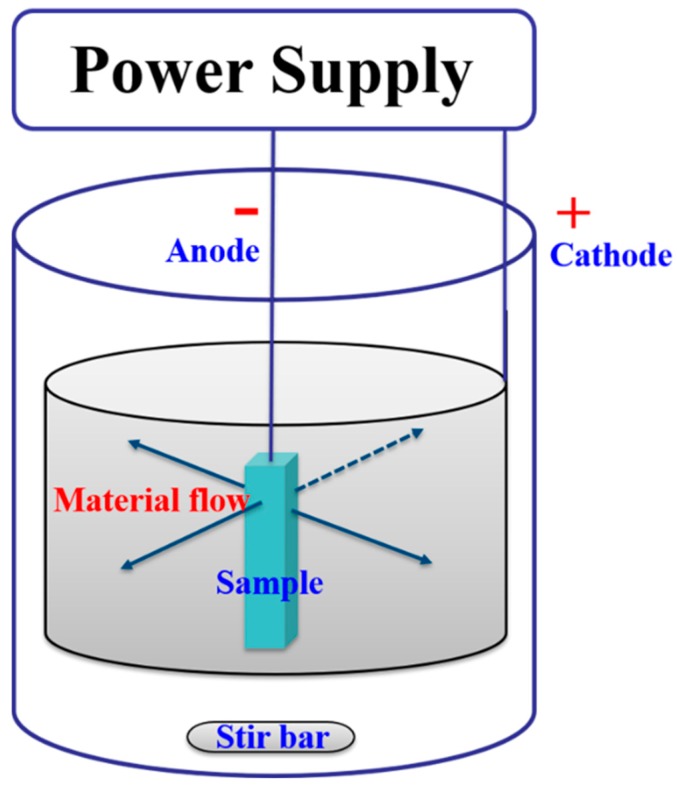
Schematic sketch of the sample and the ring-shaped cathode setup.

**Figure 3 materials-12-01466-f003:**
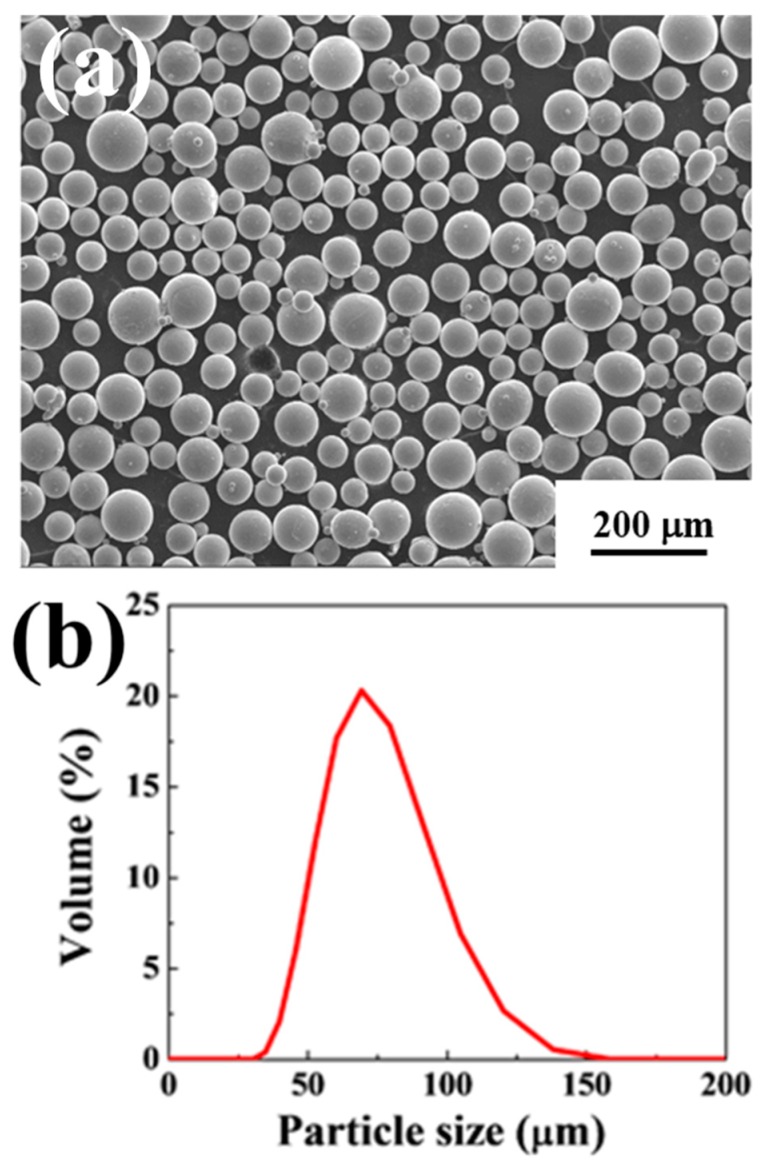
(**a**) SEM micrograph of the Ti6Al4V gas-atomized powders; and (**b**) the particle size distribution of the as-atomized powders.

**Figure 4 materials-12-01466-f004:**
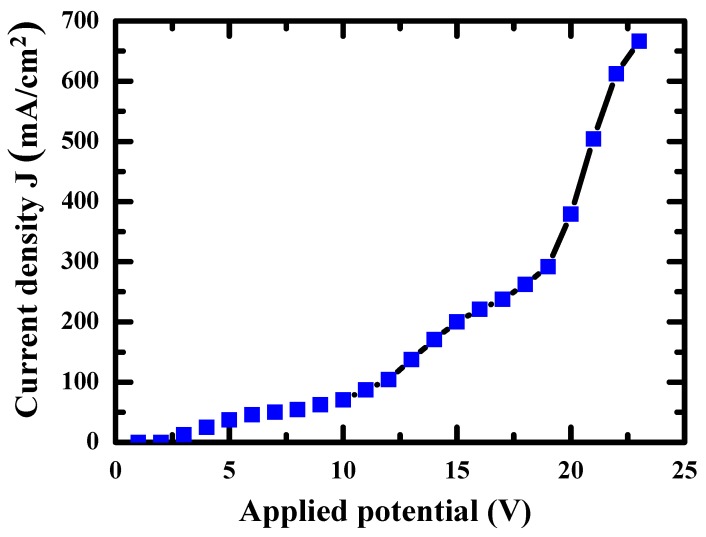
The anodic current density versus applied potential (V) plot for the EP processing.

**Figure 5 materials-12-01466-f005:**
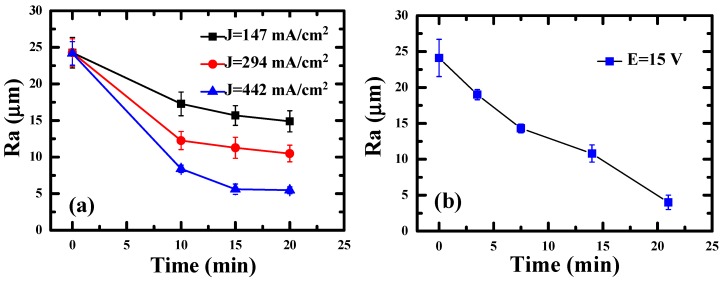
Variation of surface roughness Ra as a function of EP time: (**a**) under a constant current density of 147, 294, and 442 mA/cm^2^, respectively, and (**b**) under a constant voltage of 15 V.

**Figure 6 materials-12-01466-f006:**
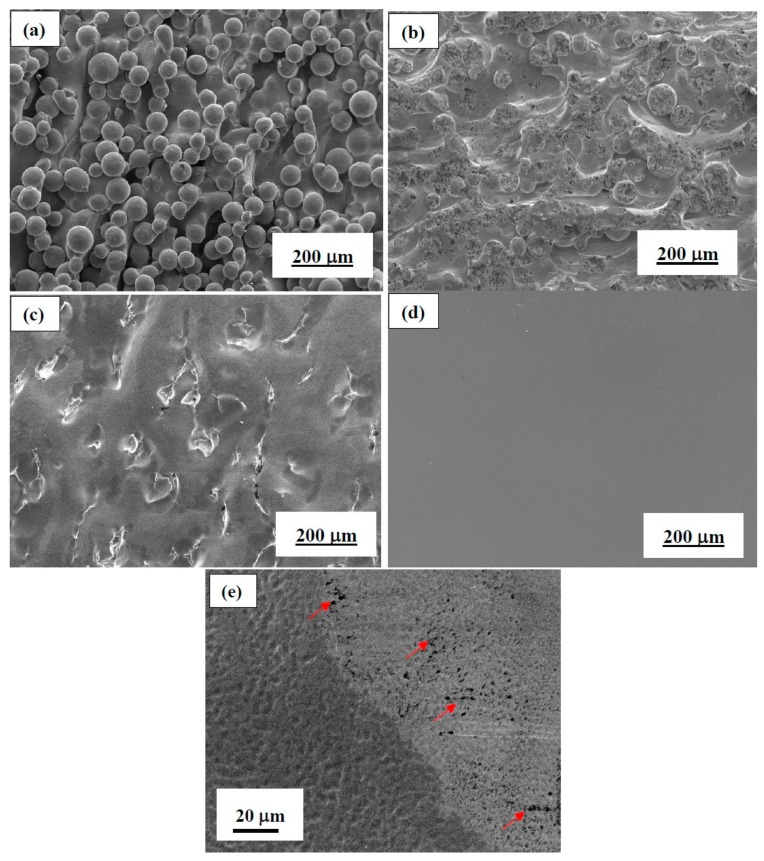
SEM micrographs of (**a**) the as-EBMed unpolished Ti6Al4V sample with Ra ~ 24 µm; and EPed samples with Ra about (**b**) 15 µm; (**c**) 10 µm; and (**d**) 4.5 µm; (**e**) Enlarged SEM micrograph showing the small black dots appeared after EP (likely over-electropolishing), and bio-corrosion pitting would be initiated from such dots, increasing the bio-corrosion current density.

**Figure 7 materials-12-01466-f007:**
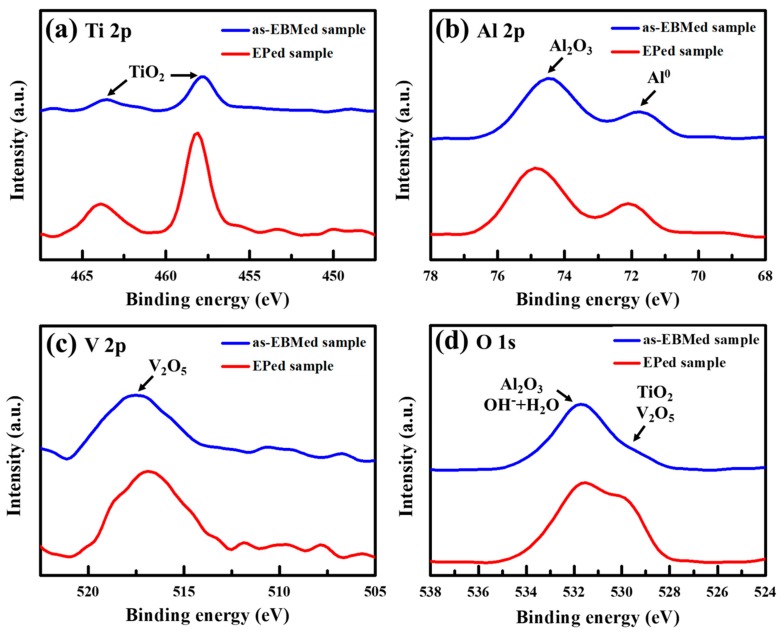
XPS spectrums of surface analysis before and after the EP: (**a**) Ti 2p; (**b**) Al 2p; (**c**) V 2p; and (**d**) O 1s peak.

**Figure 8 materials-12-01466-f008:**
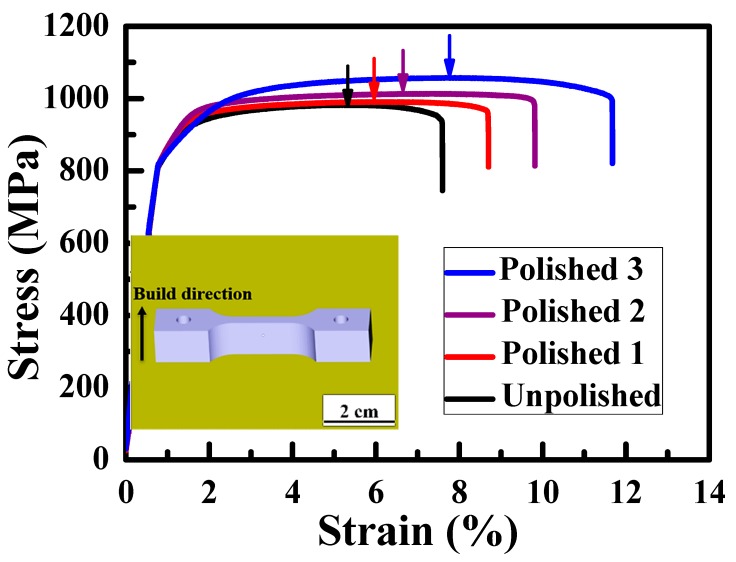
Tensile stress-strain curves of the unpolished (Ra ~ 24 µm), polished 1 (Ra ~ 15 µm), polished 2 (Ra ~ 10 µm), and polished 3 (Ra ~ 4.5 µm) samples. The necking positions were marked by arrows. The EBM printed tensile specimen designed by the CAD software and the build direction are indicated in the insert image.

**Figure 9 materials-12-01466-f009:**
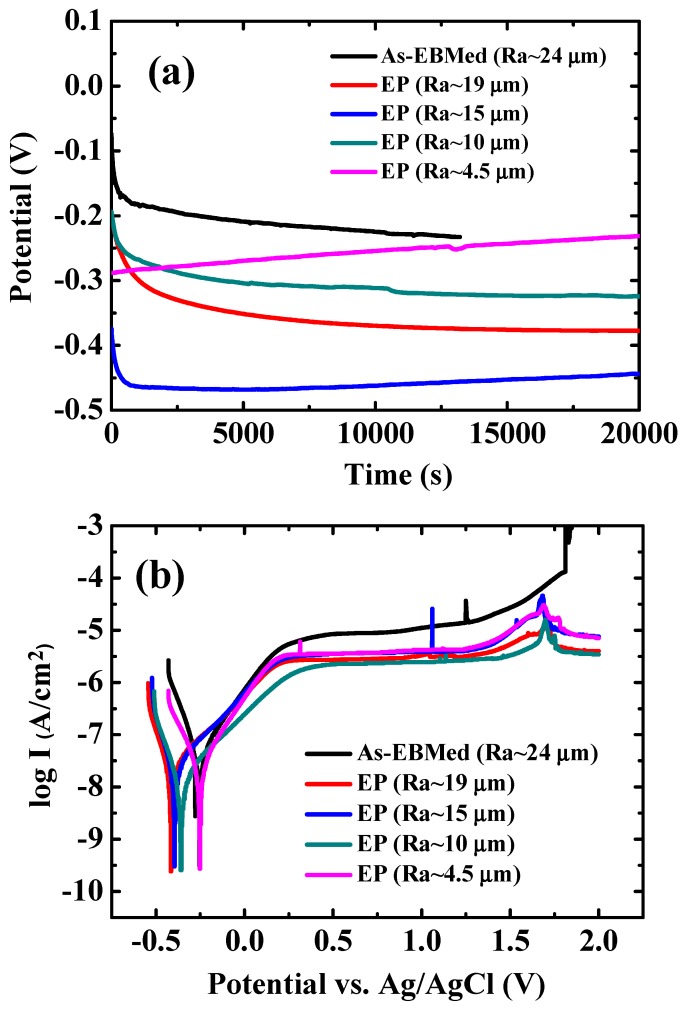
Bio-corrosion results in terms of (**a**) open circuit potential (OCP), and (**b**) potentiodynamic polarization Tafel curves of Ti6Al4V samples with different surface roughness readings, tested in simulated body fluid.

**Figure 10 materials-12-01466-f010:**
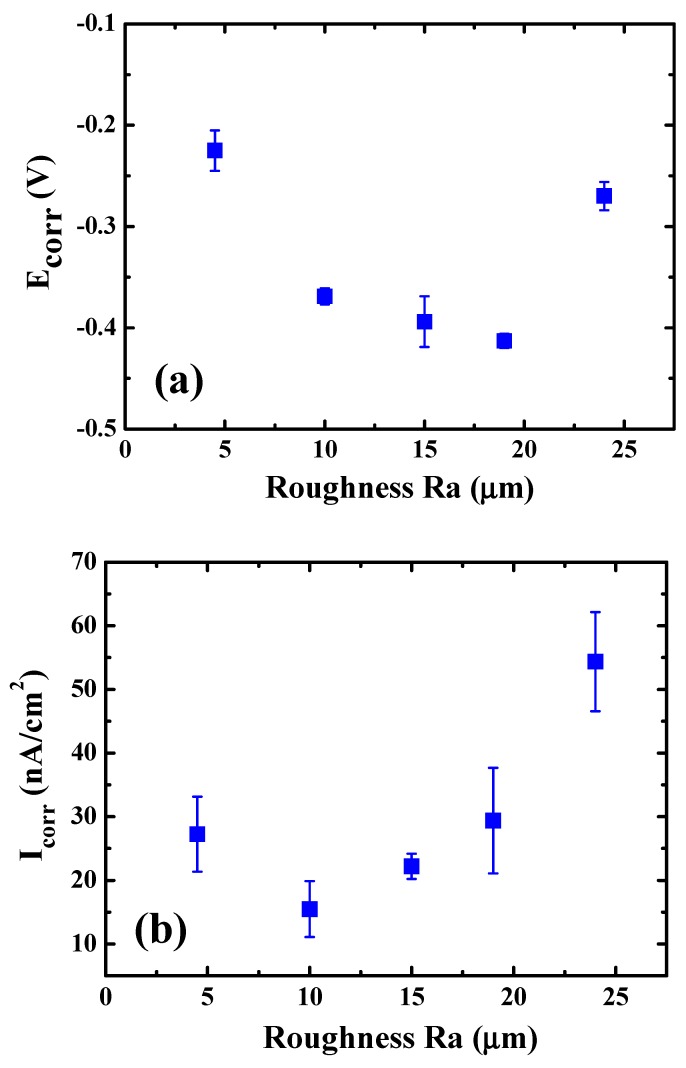
The variation of bio-corrosion (**a**) potential (E_corr_) values and (**b**) potential (I_corr_) values as a function of various surface roughness levels. The bio-corrosion activity appears to become more intense again when the roughness was smoothed by EP down to 4.5 µm, a sign of over- electropolishing.

**Table 1 materials-12-01466-t001:** Summary of physical properties of unpolished, polished 1, polished 2, and polished 3 tensile specimens. The polished 1, polished 2, and polished 3 samples are electropolished at a constant current density of 147, 294, and 442 mA/cm^2^, at 4 °C for 20 min.

	Ra (µm)	K_t_	Young’s Modulus (GPa)	Yield Stress (MPa)	Ultimate Tensile Stress (MPa)	Tensile Elongation (%)
Unpolished	24.1 ± 2.6	2.8 ± 0.3	100 ± 2	813 ± 3	995 ± 8	7.6 ± 0.4
Polished 1	15.1 ± 1.2	2.4 ± 0.1	103 ± 2	817 ± 2	1012 ± 13	8.7 ± 0.5
Polished 2	10.2 ± 0.9	2.0 ± 0.2	100 ± 1	809 ± 5	1025 ± 7	9.3 ± 0.4
Polished 3	4.5 ± 0.5	1.2 ± 0.1	102 ± 3	817 ± 2	1052 ± 8	11.6 ± 0.7

**Table 2 materials-12-01466-t002:** Summary of bio-corrosion electrochemical properties of EBM Ti6Al4V specimens with different surface roughness readings.

Ra (µm)	OCP E (V)	E_corr_ (V)	E_pit_ (V)	ΔE (V)	I_corr_ (nA/cm^2^)	I_pass_ (µA/cm^2^)
24.1 ± 2.6	−0.25	−0.270 ± 0.014	1.850 ± 0.034	>2	54.36 ± 7.80	6.71 ± 0.16
19.0 ± 0.7	−0.38	−0.413 ± 0.007	>2	>2	29.38 ± 8.28	2.80 ± 0.08
15.1 ± 1.2	−0.45	−0.394 ± 0.025	>2	>2	22.19 ± 1.97	2.88 ± 0.06
10.2 ± 1.3	−0.32	−0.369 ± 0.008	>2	>2	15.47 ± 4.40	2.68 ± 0.42
4.5 ± 0.8	−0.23	−0.225 ± 0.020	>2	>2	27.24 ± 5.89	2.90 ± 0.32

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
