# Peer review of "Effects of Electropolishing on Mechanical Properties and Bio-Corrosion of Ti6Al4V Fabricated by Electron Beam Melting Additive Manufacturing"

_materials, 2019, doi:10.3390/ma12091466_

Reviewer 1 Report

Review on

“Effects of electropolishing on mechanical properties and bio-corrosion of Ti6Al4V fabricated by electron beam melting additive manufacturing”

by Wu et al.

(manuscript materials-496337)

Submitted March 2019 to “Materials”

In this paper the authors discuss the effect of electropolishing (EP) of TiAl6V4 parts produced by electron beam melting (EBM) on mechanical and corrosion properties of the manufactured device: EP allows for a reduction of surface roughness and, thus, improvement of tensile elongation, without any negative effect on Young’s modulus or tensile strength of the part as compared to the part not treated by EP. Moreover, the authors could prove that parts properly treated by EP show higher bio-corrosion resistance, due to lower surface roughness and a denser passivated surface oxide layer as compared to the non-treated part.

The article is well-written and the addressed topic is in scope of the journal and of high interest. I only have a few points which should be addressed by the authors prior to publication.

 Detailed comments:

 Page 3, lines 109-110: why did you choose the electrolyte composition described here for EP? Was it already known to be appropriate for EP of TiAl6V? Please clarify and address this aspect in the revision.

Page 5, surface roughness: which method was used to determine Ra? In general I suggest to      add a (rather short) Materials & Methods section to mention all the used devices (e.g. type of SEM, tensile test, particle sizing equipment, XPS), the typical settings and to summarize the methods applied this work.

Author Response

Dear editor and reviewers:

Thank you very much for the review. The reviewer’s constructive comments can greatly strengthen our paper and have provided many useful suggestions. We have tried our best in replying or adding new information in text. The point to point responses are described below. 

Reviewers' comments: 

Reviewer #1:

(1) Page 3, lines 109-110: why did you choose the electrolyte composition described here for EP? Was it already known to be appropriate for EP of TiAl6V? Please clarify and  address this aspect in the revision.

Reply: Thanks very much for the constructive comment. We have tried different kinds of electrolytes for Ti6Al4V according to some previous literature. The electrolyte composition we choose in this study is the most effective one, including surface roughness improvement and uniform passive film formation. Furthermore, the electrolyte used in this study can achieve the better surface roughness after EP for a period of time. 

(2) Page 5, surface roughness: which method was used to determine Ra? In general I suggest to add a (rather short) Materials & Methods section to mention all the used devices (e.g. type of SEM, tensile test, particle sizing equipment, XPS), the typical settings and to summarize the methods applied this work.

Reply: Thank you for your suggestions. The part about Materials & Methods is shown in the Page 2-4 (lines 92-148). The 3D alpha-step profilometer (Veeco Dektak 150 Stylus Profiler) was applied to determine the surface roughness. All the specimens have been measured for at least 10 times.

Reviewer 2 Report

In this manuscript, Wu et al. extends on their earlier work on 3D printing additive manufacture and biocompatible implant-used gradient porous Ti foams, and presents experimental evidence on the effects of electropolishing on mechanical properties 2 and bio-corrosion of Ti6Al4V fabricated by electron 3 beam melting additive manufacturing. 

Electropolishing presented in this study with the lowest Ra, samples maintain the highest UTS and the highest tensile elongation. However the bio-corrosion resistance of these samples were slightly affected. They successfully showed a proper EP treatments with flat surface without tiny EP-induced defects. Overall, I found the manuscript to be well-written and is a nice follow-up work that provides additional insight on the process. I only have a couple minor points that need to be addressed, but otherwise recommend publication.

1) For Figure 5, part (a) on variation of surface roughness Ra as a function of EP time under a constant current density was missing. 

2) For the surface morphology assessment, it would be very valuable to the manuscript if the authors can provide further evaluation of different degrees of surface roughness using atomic force microscopy to better assess the uniformity and surface roughness. 

3) Please also include a short methods section that describes how the data are analyzed and which statistical tests are used. 

Author Response

Dear editor and reviewers:

Thank you very much for the review. The reviewer’s constructive comments can greatly strengthen our paper and have provided many useful suggestions. We have tried our best in replying or adding new information in text. The point to point responses are described below. 

Reviewers' comments: 

Reviewer #2:

(1) For Figure 5, part (a) on variation of surface roughness Ra as a function of EP time under a constant current density was missing.

Reply: Thank you for your comments. The following figure is part (a) of Figure 5. When the manuscript was uploaded during registration, Figure 5 was correct. However, the image of Figure 5(a) might not be displayed properly due to editing. We have revised this mistake.
(2) For the surface morphology assessment, it would be very valuable to the manuscript if the authors can provide further evaluation of different degrees of surface roughness using atomic force microscopy to better assess the uniformity and surface roughness.

Reply: Thank you very much for the constructive comment. The atomic force microscopy is usually used to determine the surface roughness, but the restriction is that it is used under nanoscale. In this study, the surface roughness is under microscale and ranges approximately from 4 mm to 24 mm. Therefore, the atomic force microscopy is not suitable for surface roughness measurement in this study. Furthermore, in order to better assess the uniformity, all the specimens including the as-EBMed and EPed ones have been measured for at least 10 times.

 (3) Please also include a short methods section that describes how the data are analyzed and which statistical tests are used.

Reply: Thanks for your constructive comments. All the raw data was collected by the devices including 3D alpha-step profilometer, universal testing machine and electrochemical analyzer system, and was exported as excel files. The software used to analysis and statistics is OriginPro 8 which is applied to draw the figures in this study. And, we can obtain the feature point from these figures. Furthermore, we have inserted this paragraph into Material and methods in the Page 4 (lines 149-152).

 Jacob C. Huang                                                 

Chair Professor, City University of Hong Kong, Hong Kong
